# Intense attosecond pulses carrying orbital angular momentum using laser plasma interactions

J.W. Wang ⬤ [1]*, M. Zepf[2,3]* & S.G. Rykovanov[4]*

Light beams with helical phase-fronts are known to carry orbital angular momentum (OAM) and provide an additional degree of freedom to beams of coherent light. While OAM beams can be readily derived from Gaussian laser beams with phase plates or gratings, this is far more challenging in the extreme ultra-violet (XUV), especially for the case of high XUV intensity. Here, we theoretically and numerically demonstrate that intense surface harmonics carrying OAM are naturally produced by the intrinsic dynamics of a relativistically intense circularly-polarized Gaussian beam (i.e. non-vortex) interacting with a target at normal incidence. Relativistic surface oscillations convert the laser pulses to intense XUV harmonic radiation via the well-known relativistic oscillating mirror mechanism. We show that the azimuthal and radial dependence of the harmonic generation process converts the spin angular momentum of the laser beam to orbital angular momentum resulting in an intense attosecond pulse (or pulse train) with OAM.

[1] State Key Laboratory of High Field Laser Physics, Shanghai Institute of Optics and Fine Mechanics, Chinese Academy of Sciences, 201800 Shanghai, China. [2] Helmholtz Institute Jena, Fröbelstieg 3, 07743 Jena, Germany. [3] Institut für Optik und Quantenelektronik, Friedrich-Schiller-Universität Jena, Max-Wien-Platz 1, 07743 Jena, Germany. [4] Center for Computational and Data-Intensive Science and Engineering, Skolkovo Institute of Science and Technology, 121205 Moscow, Russia. *email: wangjw@siom.ac.cn; m.zepf@uni-jena.de; S.Rykovanov@skoltech.ru

Angular momentum is an intrinsic property of light, with spin angular momentum (SAM) of $\pm\hbar$ per photon carried by circularly polarized (CP) light discussed by Poynting as early as 1909[1]. At the end of last century it was demonstrated that light beams with helical phase-fronts (optical vortices), described by a transverse phase structure of $\exp(-il\phi)$, where $\phi$ is the azimuthal angle, carry an OAM equivalent to $l\hbar$ per photon[2], and can therefore carry much larger OAM than is possible with SAM alone. Since then, laser beams carrying OAM have been exploited in various applications ranging from optical manipulation[3], imaging[4,5], and quantum optics[6,7], to optical communications[8,9].

Currently, OAM beams are generated by introducing azimuthally dependent phase to the initial Gaussian laser with the help of the optical elements, such as spiral phase plates[10], spatial light modulators[11], gratings[12], and q-plates[13]. XUV/X-ray pulses with OAM are of particular interest for certain classes of experiments[14,15], preferably in combination with pulse durations in the attosecond regime[16]. To date high-order harmonics in the XUV with OAM have been observed in laser-atom interactions, operating at a moderate intensity ($\sim 10^{14}$ W cm$^{-2}$) level[17–22]. With progress made in FEL beams[23], producing intense, attosecond, ultrafast XUV pulses with OAM is a challenge that has yet to be met.

The goal of achieving intense attosecond pulses with OAM in the XUV naturally suggests employing the current generation of multi-TW or even PW lasers to drive the interaction. In the relativistic regime, the proposed methods to generate intense XUV pulses with OAM mainly utilize vortex laser pulses as the drivers[24–26], which suffer from a limitation for the driving intensity. Another approach to generating OAM is spin-to-orbital

angular momentum conversion which has been demonstrated in quantum physics[27] and optics[13].

In this article, we show that the SAM of circularly polarized, high-power laser can be transferred to the harmonics carrying OAM via the relativistic oscillating mirror (ROM) mechanism, resulting in a single attosecond pulse with OAM when a few-cycle laser pulse is employed.

## Results

**Harmonics generation from a self-dented target.** The principle of SAM to OAM conversion using the ROM[28–31] mechanism is shown in Fig. 1. An intense left-handed (as defined from the point of view of the observer) circularly polarized Gaussian laser pulse impinges a plane target from the left side. The normalized peak amplitude $a_0$ of the laser field $\mathbf{E}_L$ is $a_0 = e|\mathbf{E}_L|/m_e\omega_L c = 2.8$, corresponding to a laser intensity of $1.7 \times 10^{19}$ W cm$^{-2}$. Here $m_e$ is the electron mass, $c$ is the vacuum light speed, and $-e$ is the electron charge. The laser wavelength is $\lambda_L = 0.8\,\mu$m, the laser period is $\tau = 2.67$ fs, the laser angular frequency is $\omega_L = 2.36 \times 10^{15}$ rad s$^{-1}$ and the laser spot radius is $w_0 = 4\lambda_L$. The details of simulation can be found in the Methods section. The ROM mechanism relies on the modification of the reflected waveform by the relativistic oscillation of the plasma surface. The dominant oscillation modes are at the laser frequency $\omega_L$ due to the oblique component of the laser field or at $2\omega_L$ due to the $\mathbf{v} \times \mathbf{B}$ force. This generally results in the production of both odd and even harmonic orders[29]. In the case of normal incidence only the $\mathbf{v} \times \mathbf{B}$ force remains, in which case only odd harmonics are produced

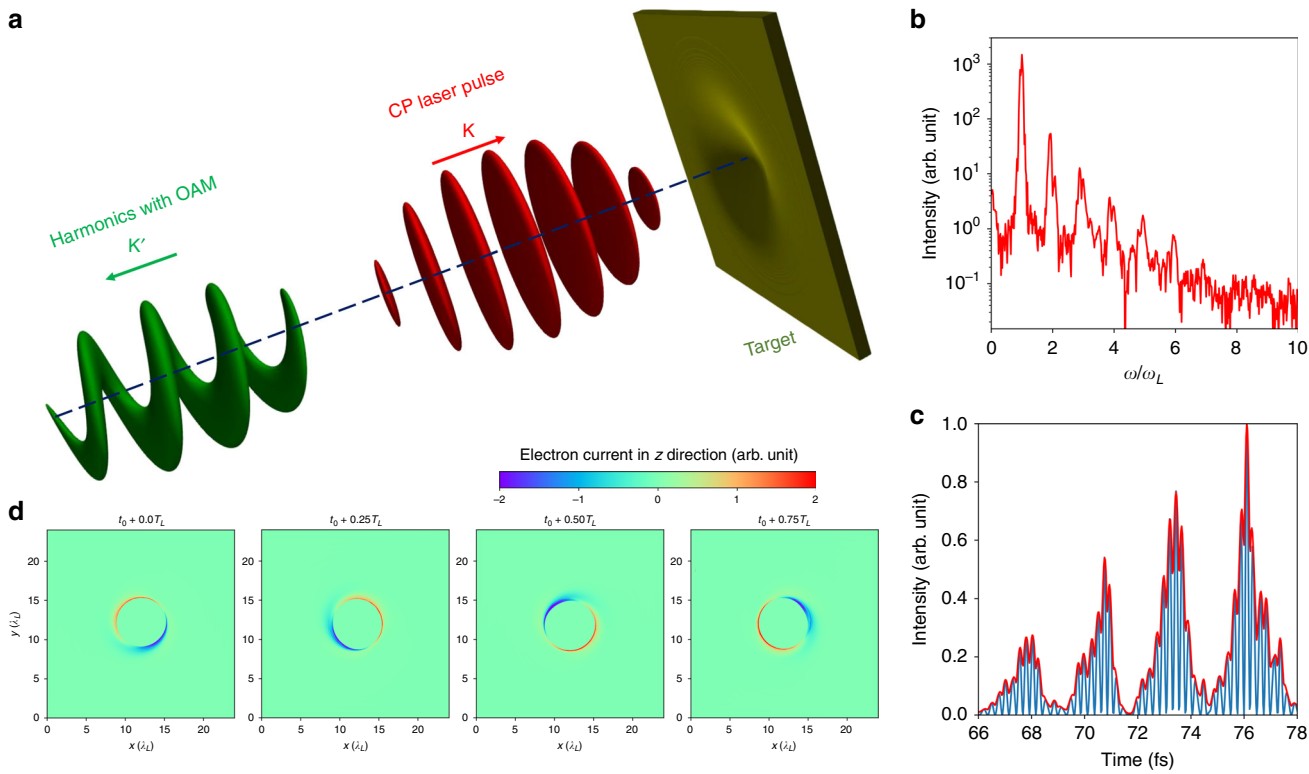

**Fig. 1** OAM harmonics generation from a self-dented target. **a** A circularly polarized Gaussian laser pulse (red) is normally incident on an initially plane target. Note that a normally incident CP Gaussian pulse does not result in harmonic generation on a plane target. However, Gaussian laser pulses deform the target by the radiation pressure of the laser pulse, resulting in electric-field component normal to off-axis laser field obliquely incident. The deformed target then transfers the spin angular momentum of the driving laser (red) to the orbital angular momentums of the reflected harmonics (green). **b** Spectrum of the off-axis reflected electric field. The frequency is normalized by the laser angular frequency $\omega_L = 2.36 \times 10^{15}$ rad s$^{-1}$. **c** An attosecond pulse train after filtering out low order ($n \leq 4$) harmonics. **d** Transverse distributions of the electron current in the $z$ direction at a fixed $z = 6.4\lambda_L$ for different times. The setup of the target can be found in the Methods section.

for linearly polarized lasers. Note that for CP pulses the $\mathbf{v} \times \mathbf{B}$ force along the laser axis is slowly varying (i.e. it follows the temporal envelope of the laser[32]). Thus, for normally incident CP Gaussian laser pulse, the $\mathbf{v} \times \mathbf{B}$ force no longer results in surface oscillations and, consequently, the reflected waveform is not modified any more and no harmonics are produced.

However, the radiation pressure along the target normal (mediated by the slowly varying $\mathbf{v} \times \mathbf{B}$ force) results in rapid target deformation (so-called denting) for sufficiently high intensities[33–36]. This breaks the symmetry of the interaction and results in the laser becoming increasingly obliquely incident away from laser axis and therefore the generation of harmonics in the outer parts of the focal spot, while the suppression of harmonic generation on the axis remains. These harmonics are driven by the perpendicular component of the transverse electric field (perpendicular to the local target surface) and result in surface oscillation with a frequency of $\omega_L$ and therefore both odd and even harmonics are expected. Our simulations of the interaction of a normally incident CP laser clearly show both odd and even harmonics in the reflected spectrum in Fig. 1b. In accordance with our expectations, the temporal structure after filtering of the lower order ($n \leq 4$) harmonics shows that a train of attosecond pulses will be obtained (Fig. 1c). The intensity of these attosecond pulses is about $2.2 \times 10^{15}\ \mathrm{W\,cm^{-2}}$, while the intensity of the driving laser pulse is $10^{19}\ \mathrm{W\,cm^{-2}}$.

**Harmonics carrying orbital angular momentum**. We recall that a characteristic of OAM beams is the vortex phase structure with zero intensity on axis. Since the harmonics in our geometry are only produced off-axis, where the laser is obliquely incident on the deformed target, the resulting intensity profile with zero emission on axis is suggestive of an OAM beam (although this is not a sufficient condition to conclude the presence of OAM). In the present scheme, the laser field component driving the oscillations can be expressed as $E_\perp = E_0 \cos\alpha \sin(\omega_L t - k_L z + \phi)$ (see Methods section), where $E_0$ is the laser field amplitude, $k_L$ is the laser vector, $\alpha$ is the opening angle of the target, and $\phi$ is the azimuthal angle of the interaction point. Clearly the oscillating phase of the plasma mirror is related with its azimuthal angle. In Fig. 1d, we plot the electron current in the $z$ direction, $J_z$, for a fixed $z$ position but at different times. The current $J_z$ is directly related to the oscillation velocity $v_z$ of the plasma mirror. We see that $J_z$ (and therefore $v_z$) is $\phi$ dependent and rotates with time. Since the harmonic emission time is directly dependent on $v_z$, the harmonic phase fronts depend on the azimuthal angle $\phi$ and surface of equal phase must meet the condition $\omega_L t - k_L z + \phi = \mathrm{const}$, corresponding to a twisted phase-front characteristic of OAM beams. In other words, OAM is introduced to the harmonics, as shown in Fig. 1a. The response of the dented target to the laser field is very similar to that of anisotropic and inhomogeneous q-plates[13], which also create OAM light beams from an incident beam carrying SAM. A detailed derivation of the OAM generation is presented in the Methods section.

Figure 2 presents the phase structure of the first three harmonic orders. As can be seen in Fig. 2a the isosurfaces have a helical structure. The number of intertwined helices depends on the order of the harmonics, e.g., the number of helices $h$ is $h = n - 1$ where $n$ is the harmonic order. The intensities distribution of each harmonic shown in the second column of Fig. 2 is doughnut-like as expected for a laser beam with OAM. The transverse distributions of the harmonic phase (third column of Fig. 2), indicates that the phases are azimuthally angle dependent. If we plot the phase along the azimuthal angle from $-\pi$ to $\pi$ for a fixed radius, we obtain sawtooth shaped functions shown in the rightmost column of Fig. 2. The number of peaks of each line corresponds to the $l$ number in the phase $\varphi \sim \exp(il\phi)$, or the

charge number $l$ in the OAM of $l\hbar$. Therefore, we can conclude that the $n$th harmonic carries $(n-1)\hbar$ OAM. A detailed analysis can be found in the Methods section. The $(n-1)\hbar$ dependence of the OAM on harmonic order $n$ can be understood simply in terms of the conservation of angular momentum. It is well known that a CP photon carries an SAM of $\hbar$ and that harmonic generation can be understood as the conversion of $n$ photons with a frequency of $\omega$ into one photon with a frequency of $n\omega$. Since the reflected harmonic field propagating to the $-z$ direction has the same polarization but opposite handedness compared to the driving field (see the calculations in Supplementary Note 1), i.e., the SAM of the emitted photon with a frequency $n\omega$ is still $\hbar$ as the incident photon, the conservation of angular momentum implies an OAM of the emitted photon of $(n-1)\hbar$.

**A single attosecond pulse with OAM**. For a target that dynamically deforms in response to the laser pressure, very few harmonics are produced in the leading edge of the laser pulse while the target is still approximately flat. However, this implies that for very short pulses, such as those required to generate a single attosecond pulse, there is insufficient time to achieve appreciable denting. We therefore consider the possibility of employing a pre-dented target. This opens the possibility of controlling efficiency and denting independent from the intensity. Importantly, it also opens up the possibility of generating a single attosecond pulse with OAM, by shooting a few-cycle and intense laser pulse on a pre-dented target[37–39]. Figure 3 presents the generation of a pulse with a duration of 450 attoseconds and with an OAM of $5\hbar$. The duration of the driving laser pulse is 5.3 fs. Since the target is predented, this short pulse can generate harmonics efficiently. We filtered from the 6th to the 16th harmonics and synthesized a single attosecond pulse. In Fig. 3a we present the structure of the single attosecond pulse, by coloring the space points with a threshold $E_x \geq 20\% E_{xmax}$, in which $E_{xmax}$ is the maximum of the filtered field $E_x$. One can see a spiral structure during $t = 6.5\tau$ and $t = 7.5\tau$, when harmonics are most efficiently generated. The transverse profile of the filtered field in Fig. 3b clearly shows its OAM with $5\hbar$. Here only a single value for the OAM is obtained because the intensity of the 6th harmonic is dominated in the filtered harmonics (see the harmonic spectrum in Supplementary Note 2). The inhomogeneity in Fig. 3b comes from the CEP effect of the few-cycle laser pulse. The temporal evolution of the filtered field at a fixed point $(x, y)$ in Fig. 3c shows the duration of the single pulse is ~450 as. The intensity of this attosecond pulse is about $10^{17}\ \mathrm{W\,cm^{-2}}$, which is two orders of magnitude higher than that of the self-denting case. While the intensities of the driving laser pulses are the same the amount $f$ of maximum denting of the targets is different ($f = 0.5\lambda_L$ for the pre-dented target and $f \approx 0.2\lambda_L$ for the self-denting case) resulting in stronger surface oscillations. What should be mentioned here is that, the pulse duration is strongly dependent on the filtering of the harmonic spectrum. Our 3D simulations are numerically limited to low orders and hence overestimate achievable pulse durations. In theory, a shorter pulse with OAM could be expected by using higher harmonic orders. For example, we present the results from 2D simulations with a higher resolution in Supplementary Note 3. It shows that up to 30th harmonic can be clearly observed and an attosecond pulse with a duration of 240 as can be obtained by filtering out the lower orders ($n \leq 15$) harmonics.

**Discussion**

In conclusion, we have found a route to converting PW class lasers to harmonics containing OAM and this is also compatible with generating an intense single attosecond pulse with OAM in the relativistic regime. The natural conversion of SAM to OAM makes

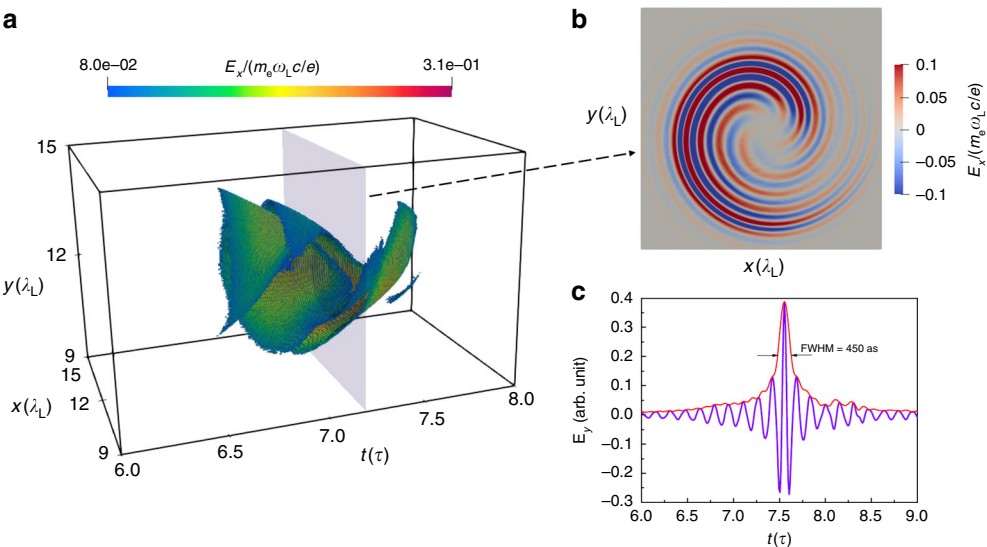

**Fig. 2** Phase structures of the harmonics. **a** Isosurface distributions of the field of each harmonic. The relative intensities for the three isosurfaces are 0.2 (red), 0.02 (yellow), 0.006 (purple), respectively. **b** Intensity distributions of each harmonic. **c** Phase distributions of each harmonic. **d** The dependence of phase on the azimuthal angle with constant radius. From the number of peaks of the serrated lines we can obtain the charge values of OAM. The first, second, and third row correspond to the 2nd, 3rd, 4th order harmonic, respectively. The profiles of the intensity and phase are obtained at a position $z = \lambda_L$, where is $5\lambda_L$ distance in front of the target.

**Fig. 3** An attosecond pulse with OAM. **a** The space structure of the instantaneous amplitude of the filtered field $E_x$. The field is filtered from the 6th to the 16th harmonics of the spectrum, when a short pulse with a FWHM of $2\tau$ is shot on a pre-dented target. In this picture only the point with a field value more than 20% of the peak is plotted, in order to demonstrate a clearer 3D structure. Here $E_x$ is normalized by $m_e\omega_L c/e(\sim 4.0 \times 10^{12} \text{ V m}^{-1})$. **b** The cross section of the filtered field (real part) at the time $7.2\tau$ which is indicated by the gray square in **a**. It shows the field carries an OAM of $5\hbar$. **c** The temporal shape of the filtered field shows the filtered reflected field is a single attosecond pulse with a duration of 450 as. The intensity of this attosecond pulse is $\sim 10^{17} \text{ W cm}^{-2}$.

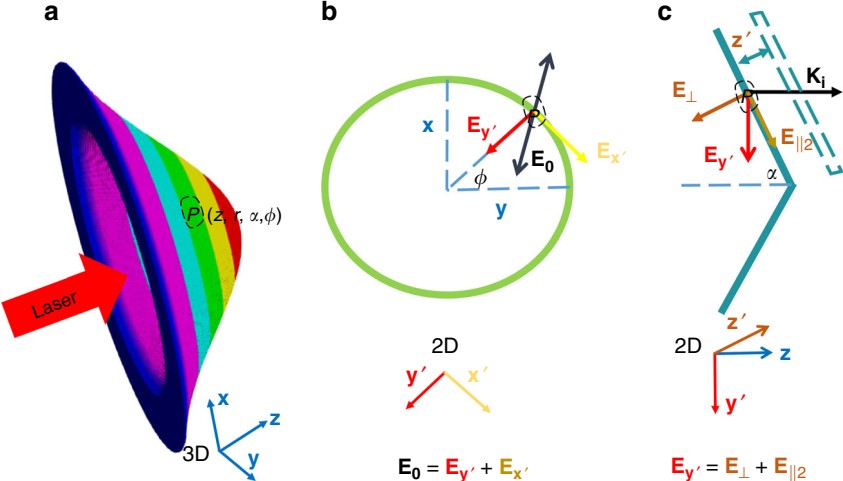

**Fig. 4** The perpendicular component of the transverse electric field. For a dented target, the component $\mathbf{E}_\perp$ of the driving electric field, which is perpendicular to the local surface, will oscillate the plasma surface. At a position $P(z, r, \alpha, \phi)$ on the target, the perpendicular component $\mathbf{E}_\perp$ can be calculated by two steps. (1) First the electric field vector $\mathbf{E_0} = E_0(r)[\sin(\omega_L t - k_L z)\mathbf{y} + \cos(\omega_L t - k_L z)\mathbf{x}]$ rotating in $\mathbf{xy}$ plane is decomposed into $\mathbf{E_{y'}} = E_0(r)\sin(\omega_L t - k_L z + \phi)\mathbf{y'}$ and $\mathbf{E_{x'}}$, $\mathbf{E_{x'}}$ is parallel to the local surface. (2) Then in $\mathbf{y'z}$ plane $\mathbf{E_{y'}}$ is decomposed again into $\mathbf{E_\perp} = \mathbf{E_{y'}}\cos\alpha = E_0(r)\cos\alpha\sin(\omega_L t - k_L z + \phi)\mathbf{z'}$ and $\mathbf{E_{\parallel 2}}$, $\mathbf{E_{\parallel 2}}$ is also parallel to the local surface.

this method more simple and more straightforward than other proposed methods[24–26] in the relativistic regime. Our results may open exciting opportunities for the applications employing intense XUV attosecond pulses carrying OAM. For example, the near-relativistic intensity of the vortex harmonics enables the applications such as transferring OAM of light to atoms[40], generating twisted gamma photons by Compton scattering[41] and manipulation of relativistic vortex cutter[42]. The attosecond duration of the isolated OAM pulse makes it an ideal tool for probing the chiral interactions[43] on the sub-femtosecond timescale and the ultrafast dynamics of spin and orbital moments in magnetic materials[44].

## Methods

**Principle of the transfer from SAM to OAM.** The harmonics generation here is interpreted by the relativistic oscillating mirror (ROM) model[28–31], in which the target (actually plasma) surface serves as a mirror to reflect the light. When the laser field drives the surface, it will simultaneously oscillate with the same frequency of the laser. The field reflected by the oscillating surface contains high-order harmonics. The reflected laser field observed at $(t, z < 0)$ was emitted at a retarded time $t_{ret} = t - Z(t_{ret})/c + z/c$ from the oscillating source, where $Z(t_{ret})$ is the position of the source at time $t_{ret}$. In the case of small curvature of the target, the electric field at the observer can be approximately expressed as[29]

$$\mathbf{e}(z, t) = \eta\mathbf{a}(t_{ret}) = \eta\mathbf{a}_0\sin(\omega_L t + k_L z - k_L Z(t)), \quad (1)$$

where $\eta$ is the reflection factor. As it turns out, it is the surface motion $Z(t)$ that generates the harmonics. Because the normally incident laser here is circularly polarized, the $\mathbf{v} \times \mathbf{B}$ force has no high frequency oscillations and therefore, it will not contribute to the harmonics generation. Only the perpendicular component $\mathbf{E}_\perp$ of the transverse electric field $\mathbf{E_0}$ will mainly oscillate the electron layer of the target surface (the longitudinal electric field coming from the tightly focusing has a weak effect here, as discussed in Supplementary Note 4). The transverse electric field can be expressed as $\mathbf{E_0} = \mathbf{E_\perp} + \mathbf{E_\parallel}$, where $\mathbf{E_\perp}$ is perpendicular to the target surface, and $\mathbf{E}$ is parallel to the target surface. From Fig. 4, one can get $\mathbf{E_\perp}$ at an arbitrary position $(z, \alpha, \phi)$ on the target surface as

$$\mathbf{E_\perp}(t, z, r, \alpha, \phi) = E_0(r)\cos\alpha\sin(\omega_L t - k_L z + \phi)\mathbf{z'}, \quad (2)$$

with $\alpha$ the open angle of the cone target, $\phi$ the azimuthal angle and $r$ the radius in the transverse profile. Driven by this field $\mathbf{E_\perp}$, the electron surface originally at $z$ moves as

$$Z'(t) = Z_s\sin(\omega_L t + \phi + \varphi_0(r)), \quad (3)$$

where $\varphi_0(r)$ is the relative phase between the driving field and the surface oscillation, and it is a function of field strength[29].

Then we can rewrite the reflected electric field in Eq. (1) as

$$\mathbf{e}(t) = \eta\mathbf{a}_0\sin(\omega_L t + \epsilon\sin(\omega_L t + \phi)), \quad (4)$$

where we dropped the relative phase $\varphi_0(r)$ and made the approximation

$Z'(t) \simeq Z(t)$. Employing the Jacobi-Anger identity[45], we can get the Fourier expansion of Eq. (4) as

$$
\begin{aligned}
e(t)/\eta a_0 &= \sum_{-\infty}^{\infty} J_n(\epsilon)\sin[(n+1)\omega_L t + n\phi] \\
&\approx \sum_0^{\infty} J_n(\epsilon)\sin[(n+1)\omega_L t + n\phi],
\end{aligned}
\quad (5)
$$

where $J_n$ is the Bessel functions of the first kind. The last step stands when $\epsilon \ll 1$. From Eq. (5) we can find that the $(n+1)$th harmonic carries an OAM with $n\hbar$.

**Particle-in-cell simulations.** The 3D simulations have been performed on JUR-ECA[46] at Jülich Supercomputing Centre and Tianhe-2 at Guangzhou National Super Computer Center using 3D Particle-in-cell (PIC) code LAPINE[47,48]. The envelope of the incident laser pulse is shaped with a function of $\sin^2(\pi t/T_0)$ $(0 \le t \le T_0)$, with $T_0 = 25\tau$ for the self-denting case and $T_0 = 4\tau$ for the pre-denting case. The target employed in the self-denting case is plane, with a thickness of 0.8 μm and a density of $2n_c$, where $n_c = 1.1 \times 10^{21}\lambda_L(\mu m)\,cm^{-3}$ is the critical density. The shape of the target surface in the pre-denting case is designed as $z = f\exp[-(x^2 + y^2)/w^2] + z_0$, with $f = 0.5\lambda_L$ and $w = 4\lambda_L$. The target is placed at $z_0 = 6\lambda_L$. A detector is placed at $z_d = \lambda_L$ to collect the reflected field. The size of the simulation box is $24\lambda_L(x) \times 24\lambda_L(y) \times 10\lambda_L(z)$ corresponding to grids $1200(x) \times 1200(y) \times 1000(z)$, with 8 macro-particles per cell. The resolution dz = $0.01\lambda_L$ will limit the order number of the harmonics that can be observed <20.

## Data availability

The data that support the findings of this study are available from the corresponding authors upon request.

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

## Acknowledgements

We are grateful to Dr. Vasily Kharin, Prof. Baifei Shen for fruitful discussions. This work was supported by the National Natural Science Foundation of China (NSFC 11674341), the Helmholtz Association (Young Investigator Group VH-NG-1037), the Strategic Priority Research Program of Chinese Academy of Sciences (Grant No. XDB1603) and Chinese Academy of Sciences President's International Fellowship Initiative (Grant No. 2018VMC0012). We gratefully acknowledge the computing time granted by the John von Neumann Institute for Computing (NIC) and provided on the supercomputer JURECA at Jülich Supercomputing Centre (JSC), and by the National Supercomputer Center in Guangzhou. Part of the simulations was performed using the Skoltech CDISE super-computer "Zhores"[49].

## Author contributions

J.W.W. and S.G.R. designed the concept. J.W.W. and S.G.R. carried out the simulations and calculations, and J.W.W. drafted the manuscript. S.G.R. and M.Z. contributed to analysing of the results and writing of the manuscript. All authors discussed the results and commented on the manuscript.

## Competing interests

The authors declare no competing interests.
