## [Peer Review File · Nature Communications]

Reviewers' comments:

Reviewer #1 (Remarks to the Author):

The revised manuscript presented by Wang et. al entitled "Intense Attosecond Pulses Carrying Orbital Angular Momentum" has been carefully revised according to the comments of the referees, and the manuscript is much improved from its initial state. I do appreciate the authors taking the comments seriously and addressing all aspects of the paper that required improvement. They have done a very nice job improving the manuscript, as well as presenting the data in well-labeled and clear figures. This is now, in my opinion, a manuscript appropriate for publication in Nature Communications. I have only a few minor additional comments, which I would encourage the authors to consider. However, the manuscript is now appropriate for publication in Nature Communications in its current state.

1. In regards to the similar arXiv publication, I agree with the authors in that an unrefereed "publication" should not be considered when addressing concerns of novelty. However, the striking similarities did, in my opinion, deserve comment, as a comparison was necessary to extract the differences between the works, which has helped to improve the strength of the manuscript. I sincerely appreciate the author's detailed response, as I have learned quite a lot from their discussion.

2. I still would have liked to see a spectrum for the pre-dented target case presented in Figure 3. Although the arguments of the authors are convincing, without seeing the amplitudes of individual harmonics it is still difficult for me to believe that the attosecond pulse obtained truly possess an OAM that is directly equal to 5. Even if it dominates the spectrum, there still should be contributions from other harmonic orders and, at best, the average OAM would be equal to 5 (and actually, slightly larger due to the higher orders).

a. Also, if the harmonics obey a power law then wouldn't the relative intensity of the H7 to H6 would be $36/49$ or $\sim 73\%$ (as so on for higher harmonics). Should such a similar amplitude contribute to the OAM value of the attosecond pulse?

3. I appreciate the authors comments on the Fourier filtering of the emitted harmonic spectrum. I had not realized that the harmonics emitted from a solid target follow a power law scaling in their intensity (which is quite different from high harmonic generation in gases), which can agree with the author obtaining an isolated attosecond pulse for the pre-dented case. However, I am surprised that the attosecond pulse is near transform limited, when considering the period of the 6th harmonics (~ 445 as). I would think that the presence of high orders, even with power law scaling, would lead some sort of chirp in the attosecond pulse.

4. Regarding the references to the literature, the authors have done a very nice job sampling the relevant, and up-to-date literature. However, it was not my intention to have the Zürich reference removed entirely. In my opinion, it should still be included as it is the seminal work of generating harmonics with OAM, and the explanations in the manuscript were given as best as they could have been with the obtained data. Although the conclusions were ultimately proved to be misguided, it still deserves citation. My only concern was it being the only reference in the original version of this manuscript, which could mislead potential readers.

Reviewer #2 (Remarks to the Author):

This is a resubmission of a manuscript I already reviewed for Nature Photonics.

Reviewing the changes and replies to all reviewers comments I believe the manuscript is now ready to be published almost as is.

I have only one minor comment at this point:

One of the referees mention the existence of an arXiv manuscript with a similar subject

(arXiv:1812.10255) and asks the authors to clarify the relation between the papers. The authors explain that while their work used the “perpendicular” component of the electric field to induce the required spin-orbit coupling, the arXiv paper used the “longitudinal” component. I must say this might be confusing, as some readers might think that both “perpendicular to the surface” and “longitudinal” with respect to the propagation direction are the same (which would be true for a flat surface). The authors better use the terms “longitudinal” and “transverse” to distinguish between the works. They should also reference the arXiv manuscript.

Reviewer #3 (Remarks to the Author):

The authors have improved the paper as compared to the previous version submitted to Nature Photonics, responding to my requests and those of the other reviewers. I am thus favorable to publishing this paper in Nature Communications.

There remain however a couple of minor technical issues that should be considered before the paper can be accepted:

1) The authors now report that the reflected harmonics have the same circular polarization as the input one, specifying that the polarization handedness is the same. I think this is incorrect (or at least reported using incorrect language). When light is normally reflected on a mirror (even without harmonics), the polarization handedness relative to propagation (= helicity) is actually flipped, that is left-handed circular polarization becomes right-handed and vice versa. This is required by conservation of angular momentum, as the outgoing photons actually have the SAME spin angular momentum orientation as the incoming one, but their propagation direction is inverted. This leads to the reversed polarization helicity (but the same spin angular momentum, relative to a fixed lab axis).

Now, also for the generated harmonics, owing to the same arguments of angular momentum conservation, the polarization handedness of output photons should be flipped with respect to the input ones. I believe that this is indeed what the authors actually find in their simulations. However, they are probably misled by the notations into interpreting incorrectly their results and state that the polarization handedness is unchanged (or alternatively into reporting their results in a slightly incorrect language). In the method section, they report the Stokes parameters of the harmonic light for given x and y axes of the lab frame, where xyz is a left-handed Cartesian frame and z is the input light propagation direction. However, for the reflected light the propagation direction is $-z$ and not z . Therefore, the same value of S_3 (for the same x and y axes) actually means OPPOSITE polarization handedness. The authors should check and fix these issues and correct the corresponding statements in the paper and in the methods section.

In terms of OAM, this implies that also the OAM of the generated harmonics should have the same sign as the outgoing photon SAM. For example, if the input photons are left-handed polarized and hence have a spin oriented as the z axis, the output harmonic photons also have the spin and the OAM oriented as the z -axis, but propagate in the $-z$ direction. In this way, the angular momentum conservation law is fulfilled and the authors' argument used to explain why the OAM is given by $n-1$, where n is the harmonic order makes perfect sense.

2) Concerning the case of a pre-dented sample, the authors now report a value in the methods for the amount of pre-denting, as given by the symbol f . However, the reported value of f is dimensionless, while it should have dimensions of length. Probably there is just a missing λ_L symbol. Again, the authors should check this issue and correct it. Moreover, I think that the amount of (maximum) pre-denting should also be reported in the main article (i.e. not only in the supplementary or methods), near to the stated values of the resulting harmonic field intensity. And it would be nice if

the authors could report also a comparison of this pre-denting with the self-induced denting taking place in the case of no pre-denting.

Response letter to Reviewer 1

The revised manuscript presented by Wang et. al entitled "Intense Attosecond Pulses Carrying Orbital Angular Momentum" has been carefully revised according to the comments of the referees, and the manuscript is much improved from its initial state. I do appreciate the authors taking the comments seriously and addressing all aspects of the paper that required improvement. They have done a very nice job improving the manuscript, as well as presenting the data in well-labeled and clear figures. This is now, in my opinion, a manuscript appropriate for publication in Nature Communications. I have only a few minor additional comments, which I would encourage the authors to consider. However, the manuscript is now appropriate for publication in Nature Communications in its current state.

We appreciate the referee's constructive suggestions to make this paper better.

1. In regards to the similar arXiv publication, I agree with the authors in that an unrefereed "publication" should not be considered when addressing concerns of novelty. However, the striking similarities did, in my opinion, deserve comment, as a comparison was necessary to extract the differences between the works, which has helped to improve the strength of the manuscript. I sincerely appreciate the author's detailed response, as I have learned quite a lot from their discussion.

In the revised version, we add a note in the supplementary information to discuss about the difference of our work with the arXiv paper.

2. I still would have liked to see a spectrum for the pre-dented target case presented in Figure 3. Although the arguments of the authors are convincing, without seeing the amplitudes of individual harmonics it is still difficult for me to believe that the attosecond pulse obtained truly possess an OAM that is directly equal to 5. Even if it dominates the spectrum, there still should be contributions from other harmonic orders and, at best, the average OAM would be equal to 5 (and actually, slightly larger due to the higher orders).

a. Also, if the harmonics obey a power law then wouldn't the relative intensity of the H7 to H6 would be $36/49$ or $\sim 73\%$ (as so on for higher harmonics). Should such a similar amplitude contribute to the OAM value of the attosecond pulse?

Although from the 1D theory the intensity of the harmonic spectrum rolls off according to the power law \$1/n^q\$ with \$q \sim 5/2\$ (G. D. Tsakiris et al., New Journal of Physics 8, 19 (2006)), the factor \$q\$ is dependent on the driving laser duration and the laser intensity (S Gordienko et al., PRL 93, 115002). The spectrum for

the pre-denting case shown in Fig. R1 indicates that $q \sim 5$ in our simulation (with $a \sim 2$, laser duration 5.3fs). In this case the intensity of the 7th harmonic is less than 50% of the 6th harmonic. We think this is why we only see OAM=5h_bar for the filtered field from 6th to 16th harmonics.

We have put the spectrum in the supplementary information to avoid overloading Figure 3.

Fig. R1. Spectrum for the pre-denting case in 3D simulation.

3. I appreciate the authors comments on the Fourier filtering of the emitted harmonic spectrum. I had not realized that the harmonics emitted from a solid target follow a power law scaling in their intensity (which is quite different from high harmonic generation in gases), which can agree with the author obtaining an isolated attosecond pulse for the pre-dented case. However, I am surprised that the attosecond pulse is near transform limited, when considering the period of the 6th harmonics (~ 445 as). I would think that the presence of high orders, even with power law scaling, would lead some sort of chirp in the attosecond pulse.

For surface harmonics generated by ROM mechanism, they are very close to Fourier transform limited, i.e. the emission of the different frequency components occurs at the same time (S. G. Rykovanov et al., New J. Phys. 10, 025025 (2008), Y. Nomura et al., Nature Physics 5, 124 (2009)).

The underlying physics of ROM can be understood in terms of relativistic Doppler effect: frequency upshifting of an EM field reflected by an electron surface moving at velocities close to the speed of light. The emission occurs in the neighborhood of the so-called relativistic γ -spikes (T. Baeva et al., Phys. Rev. E 74, 046404 (2006)), the width of which varies as $1/\gamma_{max}$, where the Lorentz factor $\gamma_{max} \sim a_L$ for $a_L > 1$. At high laser intensities high values of γ_{max} are achieved and these γ -spikes tend to be very narrow in time and symmetric so that no appreciable chirp contribution arises. Therefore, in case of ROM

harmonics, the harmonic chirp is negligible. This is in contrast to HHG from gases where each harmonic is associated with a quantum path with different return times and different atomic phases.

Note that there are effects such as surface motion that can lead to inter-cycle chirp (non-periodicity) (P. Heissler et al., PRL 108, 235003 (2012)), but these aren't relevant for the duration or chirping of an individual radiation burst.

4. Regarding the references to the literature, the authors have done a very nice job sampling the relevant, and up-to-date literature. However, it was not my intention to have the Zürich reference removed entirely. In my opinion, it should still be included as it is the seminal work of generating harmonics with OAM, and the explanations in the manuscript were given as best as they could have been with the obtained data. Although the conclusions were ultimately proved to be misguided, it still deserves citation. My only concern was it being the only reference in the original version of this manuscript, which could mislead potential readers.

In the revised version, we have added Zürich's work.

Response letter to Reviewer 2

This is a resubmission of a manuscript I already reviewed for Nature Photonics. Reviewing the changes and replies to all reviewers comments I believe the manuscript is now ready to be published almost as is.

We appreciate the referee's constructive suggestions to make this paper better.

I have only one minor comment at this point:

One of the referees mention the existence of an arXiv manuscript with a similar subject (arXiv:1812.10255) and asks the authors to clarify the relation between the papers. The authors explain that while their work used the "perpendicular" component of the electric field to induce the required spin-orbit coupling, the arXiv paper used the "longitudinal" component. I must say this might be confusing, as some readers might think that both "perpendicular to the surface" and "longitudinal" with respect to the propagation direction are the same (which would be true for a flat surface). The authors better use the terms "longitudinal" and "transverse" to distinguish between the works. They should also reference the arXiv manuscript.

In the revised version, we add a note in the supplementary information to discuss about the difference of our work with the arXiv paper. And the arXiv paper is cited.

Response letter to Reviewer 3

The authors have improved the paper as compared to the previous version submitted to Nature Photonics, responding to my requests and those of the other reviewers. I am thus favorable to publishing this paper in Nature Communications.

We appreciate the referee's constructive suggestions to make this paper better.

There remain however a couple of minor technical issues that should be considered before the paper can be accepted:

1) The authors now report that the reflected harmonics have the same circular polarization as the input one, specifying that the polarization handedness is the same. I think this is incorrect (or at least reported using incorrect language). When light is normally reflected on a mirror (even without harmonics), the polarization handedness relative to propagation (= helicity) is actually flipped, that is left-handed circular polarization becomes right-handed and vice versa. This is required by conservation of angular momentum, as the outgoing photons actually have the SAME spin angular momentum orientation as the incoming one, but their propagation direction is inverted. This leads to the reversed polarization helicity (but the same spin angular momentum, relative to a fixed lab axis).

Now, also for the generated harmonics, owing to the same arguments of angular momentum conservation, the polarization handedness of output photons should be flipped with respect to the input ones. I believe that this is indeed what the authors actually find in their simulations. However, they are probably misled by the notations into interpreting incorrectly their results and state that the polarization handedness is unchanged (or alternatively into reporting their results in a slightly incorrect language). In the method section, they report the Stokes parameters of the harmonic light for given x and y axes of the lab frame, where xyz is a left-handed Cartesian frame and z is the input light propagation direction. However, for the reflected light the propagation direction is $-z$ and not z . Therefore, the same value of S_3 (for the same x and y axes) actually means OPPOSITE polarization handedness. The authors should check and fix these issues and correct the corresponding statements in the paper and in the methods section.

In terms of OAM, this implies that also the OAM of the generated harmonics should have the same sign as the outgoing photon SAM. For example, if the input photons are left-handed polarized and hence have a spin oriented as the z axis, the output harmonic photons also have the spin and the OAM oriented as the z -axis, but propagate in the $-z$ direction. In this way, the angular momentum conservation law is fulfilled and the authors' argument used to explain why the OAM is given by $n-1$, where n is the harmonic order makes

perfect sense.

We appreciate the referee for pointing out our mistakes. We had not realized that the harmonics are propagating to $-z$ direction when calculated the Stokes parameters. We have then carefully checked our calculations again, and found that the reflected field indeed should be right-handed (as defined from the point of view of the observer), opposite handedness compared to the incident field. We have corrected the statements in page 6 of the revised paper and page 2 of the Supplementary Information.

2) Concerning the case of a pre-dented sample, the authors now report a value in the methods for the amount of pre-denting, as given by the symbol f . However, the reported value of f is dimensionless, while it should have dimensions of length. Probably there is just a missing λ_L symbol. Again, the authors should check this issue and correct it. Moreover, I think that the amount of (maximum) pre-denting should also be reported in the main article (i.e. not only in the supplementary or methods), near to the stated values of the resulting harmonic field intensity. And it would be nice if the authors could report also a comparison of this pre-denting with the self-induced denting taking place in the case of no pre-denting.

Thanks for the referee's reminding. The symbol f does have a unit of λ_L . We have corrected it in page 10. We have also presented the amounts of maximum denting for both pre-denting and self-denting cases in page 7.

REVIEWERS' COMMENTS:

Reviewer #1 (Remarks to the Author):

The revised manuscript presented by Wang et. al entitled "Intense Attosecond Pulses Carrying Orbital Angular Momentum" has now been through 3 rounds of revision and the quality of the work and presentation of the results has improved at each step. The authors have done a great job addressing the comments of the reviewers, and, in my opinion, the manuscript is now ready for publication in Nature Communications, without further modification.

Reviewer #3 (Remarks to the Author):

The authors have corrected the technical issues that I mentioned in my previous report and have answered to the comments of the other reviewers in a satisfactory way. Hence, I can now recommend publication.